# KRONECKER RECURRENT UNITS

## ABSTRACT

Our work addresses two important issues with recurrent neural networks: (1) they are over-parametrized, and (2) the recurrent weight matrix is ill-conditioned. The former increases the sample complexity of learning and the training time. The latter causes the vanishing and exploding gradient problem.

We present a flexible recurrent neural network model called Kronecker Recurrent Units (KRU). KRU achieves parameter efficiency in RNNs through a Kronecker factored recurrent matrix. It overcomes the ill-conditioning of the recurrent matrix by enforcing soft unitary constraints on the factors. Thanks to the small dimensionality of the factors, maintaining these constraints is computationally efficient.

Our experimental results on seven standard data-sets reveal that KRU can reduce the number of parameters by three orders of magnitude in the recurrent weight matrix compared to the existing recurrent models, without trading the statistical performance.

These results in particular show that while there are advantages in having a high dimensional recurrent space, the capacity of the recurrent part of the model can be dramatically reduced.

## 1 INTRODUCTION

Deep neural networks have defined the state-of-the-art in a wide range of problems in computer vision, speech analysis, and natural language processing (Krizhevsky et al., 2012; Hinton et al., 2012; Mikolov, 2012). However, these models suffer from two key issues. (1) They are over-parametrized; thus it takes a very long time for training and inference. (2) Learning deep models is difficult because of the poor conditioning of the matrices that parameterize the model. These difficulties are especially relevant to recurrent neural networks. Indeed, the number of distinct parameters in RNNs grows as the square of the size of the hidden state conversely to convolutional networks which enjoy weight sharing. Moreover, poor conditioning of the recurrent matrices results in the gradients to explode or vanish exponentially fast along the time horizon. This problem prevents RNN from capturing long-term dependencies (Hochreiter, 1991; Bengio et al., 1994).

There exists an extensive body of literature addressing over-parametrization in neural networks. LeCun et al. (1990) first studied the problem and proposed to remove unimportant weights in neural networks by exploiting the second order information. Several techniques which followed include low-rank decomposition (Denil et al., 2013), training a small network on the soft-targets predicted by a big pre-trained network (Ba & Caruana, 2014), low bit precision training (Courbariaux et al., 2014), hashing (Chen et al., 2015), etc. A notable exception is the deep fried convnets (Yang et al., 2015) which explicitly parameterizes the fully connected layers in a convnet with a computationally cheap and parameter-efficient structured linear operator, the Fastfood transform (Le et al., 2013). These techniques are primarily aimed at feed-forward fully connected networks and very few studies have focused on the particular case of recurrent networks (Arjovsky et al., 2016).

The problem of vanishing and exploding gradients has also received significant attention. Hochreiter & Schmidhuber (1997) proposed an effective gating mechanism in their seminal work on LSTMs. Later, this technique was adopted by other models such as the Gated Recurrent Units (GRU) (Chung et al., 2015) and the Highway networks (Srivastava et al., 2015) for recurrent and feed-forward neural networks respectively. Other popular strategies include gradient clipping (Pascanu et al., 2013), and orthogonal initialization of the recurrent weights (Le et al., 2015). More recently (Arjovsky et al., 2016) proposed to use a unitary recurrent weight matrix. The use of norm preserving unitary maps

prevent the gradients from exploding or vanishing, and thus help to capture long-term dependencies. The resulting model called unitary RNN (uRNN) is computationally efficient since it only explores a small subset of general unitary matrices. Unfortunately, since uRNNs can only span a reduced subset of unitary matrices their expressive power is limited (Wisdom et al., 2016). We denote this restricted capacity unitary RNN as RC uRNN. Full capacity unitary RNN (FC uRNN) (Wisdom et al., 2016) proposed to overcome this issue by parameterizing the recurrent matrix with a full dimensional unitary matrix, hence sacrificing computational efficiency. Indeed, FC uRNN requires a computationally expensive projection step which takes $\mathcal{O}(N^3)$ time ($N$ being the size of the hidden state) at each step of the stochastic optimization to maintain the unitary constraint on the recurrent matrix. Mhammedi et al. (2016) in their orthogonal RNN (oRNN) avoided the expensive projection step in FC uRNN by parametrizing the orthogonal matrices using Householder reflection vectors, it allows a fine-grained control over the number of parameters by choosing the number of Householder reflection vectors. When the number of Householder reflection vector approaches $N$ this parametrization spans the full reflection set, which is one of the disconnected subset of the full orthogonal set. Jing et al. (2017) also presented a way of parametrizing unitary matrices which allows fine-grained control on the number of parameters. This work called as Efficient Unitary RNN (EURNN), exploits the continuity of unitary set to have a tunable parametrization ranging from a subset to the full unitary set.

Although the idea of parametrizing recurrent weight matrices with strict unitary linear operator is appealing, it suffers from several issues: (1) Strict unitary constraints severely restrict the search space of the model, thus making the learning process unstable. (2) Strict unitary constraints make forgetting irrelevant information difficult. While this may not be an issue for problems with non-vanishing long term influence, it causes failure when dealing with real world problems that have vanishing long term influence 4.7. Henaff et al. (2016) have previously pointed out that the good performance of strict unitary models on certain synthetic problems is because it exploits the biases in these data-sets which favors a unitary recurrent map and these models may not generalize well to real world data-sets. More recently Vorontsov et al. (2017) have also studied this problem of unitary RNNs and the authors found out that relaxing the strict unitary constraint on the recurrent matrix to a soft unitary constraint improved the convergence speed as well as the generalization performance.

Our motivation is to address the problems of existing recurrent networks mentioned above. We present a new model called Kronecker Recurrent Units (KRU). At the heart of KRU is the use of Kronecker factored recurrent matrix which provide an elegant way to adjust the number of parameters to the problem at hand. This factorization allows us to finely modulate the number of parameters required to encode $N \times N$ matrices, from $\mathcal{O}(\log(N))$ when using factors of size $2 \times 2$, to $\mathcal{O}(N^2)$ parameters when using a single factor of the size of the matrix itself. We tackle the vanishing and exploding gradient problem through a soft unitary constraint (Jose & Fleuret, 2016; Henaff et al., 2016; Cisse et al., 2017; Vorontsov et al., 2017). Thanks to the properties of Kronecker matrices (Van Loan, 2000), this constraint can be enforced efficiently. Please note that KRU can readily be plugged into vanilla real space RNN, LSTM and other variants in place of standard recurrent matrices. However in case of LSTMs we do not need to explicitly enforce the approximate orthogonality constraints as the gating mechanism is designed to prevent vanishing and exploding gradients. Our experimental results on seven standard data-sets reveal that KRU and KRU variants of real space RNN and LSTM can reduce the number of parameters drastically (hence the training and inference time) without trading the statistical performance. Our core contribution in this work is a flexible, parameter efficient and expressive recurrent neural network model which is robust to vanishing and exploding gradient problem.

The paper is organized as follows, in section 2 we restate the formalism of RNN and detail the core motivations for KRU. In section 3 we present the Kronecker recurrent units (KRU). We present our experimental findings in section 4 and section 5 concludes our work.

Table 1: Notations

| | |
|---|---|
| $D, N, M$ | Input, hidden and output dimensions |
| $\mathbf{x}_t \in \mathbb{R}^D$ or $\mathbb{C}^D$, $\mathbf{h}_t \in \mathbb{C}^D$ | Input and hidden state at time $t$ |
| $\mathbf{y}_t \in \mathbb{R}^M$ or $\mathbb{C}^M$, $\hat{\mathbf{y}}_t \in \mathbb{R}^M$ or $\mathbb{C}^M$ | Prediction targets and RNN predictions at time $t$ |
| $\mathbf{U} \in \mathbb{C}^{N \times D}, \mathbf{W} \in \mathbb{C}^{N \times N}, \mathbf{V} \in \mathbb{C}^{M \times N}$ | Input, recurrent amd output weight matrices |
| $\mathbf{b} \in \mathbb{R}^N$ or $\mathbb{C}^N, \mathbf{c} \in \mathbb{R}^M$ or $\mathbb{C}^M$ | Hidden and output bias |
| $\sigma(.), \mathcal{L}(\hat{\mathbf{y}}, \mathbf{y})$ | Point-wise non-linear activation function and the loss function |

## 2 RECURRENT NEURAL NETWORK FORMALISM

Table 1 summarizes some notations that we use in the paper. We consider the field to be complex rather than real numbers. We will motivate the choice of complex numbers later in this section. Consider a standard recurrent neural network (Elman, 1990). Given a sequence of $T$ input vectors: $\mathbf{x}_0, \mathbf{x}_1, \ldots, \mathbf{x}_{T-1}$, at a time step $t$ RNN performs the following:

$$\mathbf{h}_t = \sigma(\mathbf{W}\mathbf{h}_{t-1} + \mathbf{U}\mathbf{x}_t + \mathbf{b}) \tag{1}$$
$$\hat{\mathbf{y}}_t = \mathbf{V}\mathbf{h}_t + \mathbf{c}, \tag{2}$$

where $\hat{\mathbf{y}}_t$ is the predicted value at time step $t$.

### 2.1 OVER PARAMETERIZATION AND COMPUTATIONAL EFFICIENCY

The total number of parameters in a RNN is $c(DN + N^2 + N + M + MN)$, where $c$ is 1 for real and 2 for complex parametrization. As we can see, the number of parameters grows quadratically with the hidden dimension, $i.e.$, $\mathcal{O}(N^2)$. We show in the experiments that this quadratic growth is an over parametrization for many real world problems. Moreover, it has a direct impact on the computational efficiency of RNNs because the evaluation of $\mathbf{W}\mathbf{h}_{t-1}$ takes $\mathcal{O}(N^2)$ time and it recursively depends on previous hidden states. However, other components $\mathbf{U}\mathbf{x}_t$ and $\mathbf{V}\mathbf{h}_t$ can usually be computed efficiently by a single matrix-matrix multiplication for each of the components. That is, we can perform $\mathbf{U}[\mathbf{x}_0, \ldots, \mathbf{x}_T]$ and $\mathbf{V}[\mathbf{h}_0, \ldots, \mathbf{h}_{T-1}]$, this is efficient using modern BLAS libraries. So to summarize, if we can control the number of parameters in the recurrent matrix $\mathbf{W}$, then we can control the computational efficiency.

### 2.2 POOR CONDITIONING IMPLIES GRADIENTS EXPLODE OR VANISH

The vanishing and exploding gradient problem refers to the decay or growth of the partial derivative of the loss $\mathcal{L}(.)$ with respect to the hidden state $\mathbf{h}_t$ $i.e.$ $\frac{\partial \mathcal{L}}{\partial \mathbf{h}_t}$ as the number of time steps $T$ grows (Arjovsky et al., 2016). By the application of the chain rule, the following can be shown (Arjovsky et al., 2016):

$$\left\| \frac{\partial \mathcal{L}}{\partial \mathbf{h}_t} \right\| \leq \|\mathbf{W}\|^{T-t}. \tag{3}$$

From Equation 3, it is clear that if the absolute value of the eigenvalues of $\mathbf{W}$ deviates from 1 then $\frac{\partial \mathcal{L}}{\partial \mathbf{h}_t}$ may explode or vanish exponentially fast with respect to $T - t$. So a strategy to prevent vanishing and exploding gradient is to control the spectrum of $\mathbf{W}$.

### 2.3 WHY COMPLEX FIELD?

Although Arjovsky et al. (2016) and Wisdom et al. (2016) use complex valued networks with unitary constraints on the recurrent matrix, the motivations for such models are not clear. We give a simple but compelling reason for complex-valued recurrent networks.

The absolute value of the determinant of a unitary matrix is 1. Hence in the real space, the set of all unitary (orthogonal) matrices have a determinant of 1 or −1, $i.e.$, the set of all rotations and reflections respectively. Since the determinant is a continuous function, the unitary set in real space is disconnected. Consequently, with the real-valued networks we cannot span the full unitary set using the standard continuous optimization procedures. On the contrary, the unitary set is connected in the complex space as its determinants are the points on the unit circle and we do not have this issue.

As we mentioned in the introduction (Jing et al., 2017) uses this continuity of unitary space to have a tunable continuous parametrization ranging from subspace to full unitary space. Any continuous parametrization in real space can only span a subset of the full orthogonal set. For example, the Householder parametrization (Mhammedi et al., 2016) suffers from this issue.

## 3 KRONECKER RECURRENT UNITS (KRU)

We consider parameterizing the recurrent matrix $\mathbf{W}$ as a Kronecker product of $F$ matrices $\mathbf{W}_0, \ldots, \mathbf{W}_{F-1}$,

$$\mathbf{W} = \mathbf{W}_0 \otimes \cdots \otimes \mathbf{W}_{F-1} = \otimes_{f=0}^{F-1} \mathbf{W}_f. \tag{4}$$

Where each $\mathbf{W}_f \in \mathbb{C}^{P_f \times Q_f}$ and $\prod_{f=0}^{F-1} P_f = \prod_{f=0}^{F-1} Q_f = N$. $\mathbf{W}_f$'s are called as Kronecker factors.

To illustrate the Kronecker product of matrices, let us consider the simple case when $\forall_f \{ P_f = Q_f = 2 \}$. This implies $F = \log_2 N$. And $\mathbf{W}$ is recursevly defined as follows:

$$\mathbf{W} = \otimes_{f=0}^{\log_2 N - 1} \mathbf{W}_f = \begin{bmatrix} \mathbf{w}_0(0,0) & \mathbf{w}_0(0,1) \\ \mathbf{w}_0(1,0) & \mathbf{w}_0(1,1) \end{bmatrix} \otimes_{f=1}^{\log_2 N - 1} \mathbf{W}_f, \tag{5}$$

$$= \begin{bmatrix} \mathbf{w}_0(0,0)\mathbf{W}_1 & \mathbf{w}_0(0,1)\mathbf{W}_1 \\ \mathbf{w}_0(1,0)\mathbf{W}_1 & \mathbf{w}_0(1,1)\mathbf{W}_1 \end{bmatrix} \otimes_{f=2}^{\log_2 N - 1} \mathbf{W}_f. \tag{6}$$

When $\forall_f \{ p_f = q_f = 2 \}$ the number of parameters is $8 \log_2 N$ and the time complexity of hidden state computation is $\mathcal{O}(N \log_2 N)$. When $\forall_f \{ p_f = q_f = N \}$ then $F = 1$ and we will recover standard complex valued recurrent neural network. We can span every Kronecker representations in between by choosing the number of factors and the size of each factor. In other words, the number of Kronecker factors and the size of each factor give us fine-grained control over the number of parameters and hence over the computational efficiency. This strategy allows us to design models with the appropriate trade-off between computational budget and statistical performance. All the existing models lack this flexibility.

The idea of using Kronecker factorization for approximating Fisher matrix in the context of natutal gradient methods have recently recieved much attention. The algorithm was originally presented in Martens & Grosse (2015) and was later extended to convolutional layers (Grosse & Martens, 2016), distributed second order optimization (Ba et al., 2016) and for deep reinforcement learning (Wu et al., 2017). However Kronecker matrices have not been well explored as learnable parameters except (Zhang et al., 2015) used it's spectral property for fast orthogonal projection and (Zhou et al., 2015) used it as a layer in convolutional neural networks.

### 3.1 SOFT UNITARY CONSTRAINT

Poor conditioning results in vanishing or exploding gradients. Unfortunately, the standard solution which consists of optimization on the strict unitary set suffers from the retention of noise over time. Indeed, the small eigenvalues of the recurrent matrix can represent a truly vanishing long-term influence on the particular problem and in that sense, there can be good or bad vanishing gradients. Consequently, enforcing strict unitary constraint (forcing the network to never forget) can be a bad strategy. A simple solution to get the best of both worlds is to enforce unitary constraint approximately by using the following regularization:

$$\left\| \mathbf{W}_f^H \mathbf{W}_f - \mathbf{I} \right\|^2, \forall f \in \{0, \ldots, F-1\} \tag{7}$$

Please note that these constraints are enforced on each factor of the Kronecker factored recurrent matrix. This procedure is computationally very efficient since the size of each factor is typically small. It suffices to do so because if each of the Kronecker factors $\{\mathbf{W}_0, \ldots, \mathbf{W}_{F-1}\}$ are unitary then the full matrix $\mathbf{W}$ is unitary (Van Loan, 2000) and if each of the factors are approximately unitary then the full matrix is approximately unitary. We apply soft unitary constraints as a regularizer whose strength is cross-validated on the validation set.

This type of regularizer has recently been exploited for real-valued models. (Cisse et al., 2017) showed that enforcing approximate orthogonality constraint on the weight matrices make the network robust to adversarial samples as well as improve the learning speed. In metric learning (Jose & Fleuret, 2016) have shown that it better conditions the projection matrix thereby improving the robustness of stochastic gradient over a wide range of step sizes as well asthe generalization performance. Henaff et al. (2016) and Vorontsov et al. (2017) have also used this soft unitary contraints on standard RNN after identifying the problems with the strict unitary RNN models. However the computational complexity of naively applying this soft constraint is $\mathcal{O}(N^3)$. This is prohibitive for RNNs with large hidden state unless one considers a Kronecker factorization.

## 4 EXPERIMENTS

Existing deep learning libraries such as Theano (Bergstra et al., 2011), Tensorflow (Abadi et al., 2016) and Pytorch (Paszke et al., 2017) do not support fast primitives for Kronecker products with arbitrary number of factors. So we wrote custom CUDA kernels for Kronecker forward and backward operations. All our models are implemented in C++. We will release our library to reproduce all the results which we report in this paper. We use tanh as activation function for RNN, LSTM and our model KRU-LSTM. Whereas RC uRNN, FC uRNN and KRU uses complex rectified linear units (Arjovsky et al., 2016).

### 4.1 COPY MEMORY PROBLEM

Copy memory problem (Hochreiter & Schmidhuber, 1997) tests the model's ability to recall a sequence after a long time gap. In this problem each sequence is of length $T + 20$ and each element in the sequence come from 10 classes $\{0, \ldots, 9\}$. The first 10 elements are sampled uniformly with replacement from $\{1, \ldots, 8\}$. The next $T - 1$ elements are filled with 0, the 'blank' class followed by 9, the 'delimiter' and the remaining 10 elements are 'blank' category. The goal of the model is to output a sequence of $T + 10$ blank categories followed by the 10 element sequence from the beginning of the input sequence. The expected average cross entropy for a memory-less strategy is $\frac{10 \log 8}{T+20}$.

Our experimental setup closely follows Wisdom et al. (2016) which in turn follows Arjovsky et al. (2016) but $T$ extended to 1000 and 2000. Our model, KRU uses a hidden dimension $N$ of 128 with 2x2 Kronecker factors which corresponds to $\approx$5K parameters in total. We use a RNN of $N = 128$ ($\approx$ 19K parameters) , LSTM of $N = 128$ ( $\approx$ 72K parameters), RC uRNN of $N = 470$ ( $\approx$ 21K parameters) , FC uRNN of $N = 128$ ( $\approx$ 37K parameters). All the baseline models are deliberately chosen to have more parameters than KRU. Following Wisdom et al. (2016); Arjovsky et al. (2016), we choose the training and test set size to be 100K and 10K respectively. All the models were trained using RMSprop with a learning rate of $1e-3$, decay of 0.9 and a batch size of 20. For both the settings $T = 1000$ and $T = 2000$, KRU converges to zero average cross entropy faster than FC uRNN. All the other baselines are stuck at the memory-less cross entropy.

The results are shown in figure 1. For this problem we do not learn the recurrent matrix of KRU, We initialize it by random unitary matrix and just learn the input to hidden, hidden to output matrices and the bias. We found out that this strategy already solves the problem faster than all other methods. Our model in this case is similar to a parametrized echo state networks (ESN). ESNs are known to be able to learn long-term dependencies if they are properly initialized (Jaeger, 2001). We argue that this data-set is not an ideal benchmark for evaluating RNNs in capturing long term dependencies. Just a unitary initialization of the recurrent matrix would solve the problem.

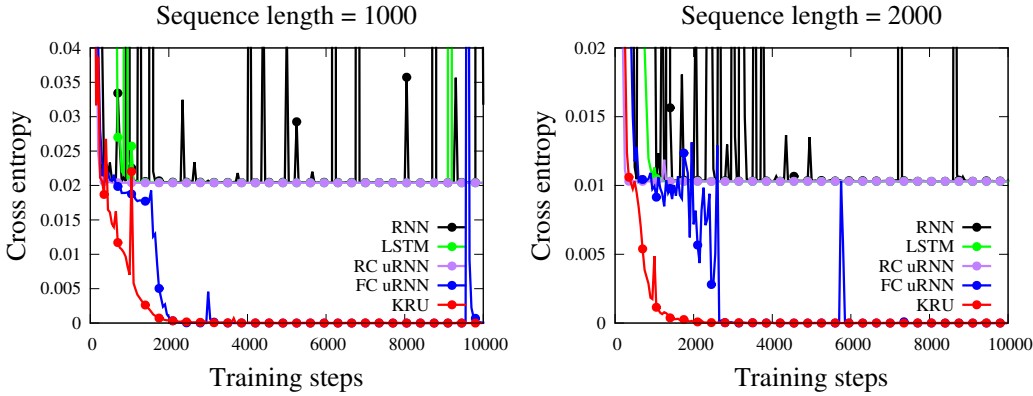

Figure 1: Learning curves on copy memory problem for $T$=1000 and $T$=2000.

## 4.2 ADDING PROBLEM

Following Arjovsky et al. (2016) we describe the adding problem (Hochreiter & Schmidhuber, 1997). Each input vector is composed of two sequences of length $T$. The first sequence is sampled from $\mathcal{U}[0, 1]$. In the second sequence exactly two of the entries is 1, the 'marker' and the remaining is 0. The first 1 is located uniformly at random in the first half of the sequence and the other 1 is located again uniformly at random in the other half of the sequence. The network's goal is to predict the sum of the numbers from the first sequence corresponding to the marked locations in the second sequence.

We evaluate four settings as in Arjovsky et al. (2016) with $T$=100, $T$=200, $T$=400, and $T$=750. For all four settings, KRU uses a hidden dimension $N$ of 512 with 2x2 Kronecker factors which corresponds to ≈3K parameters in total. We use a RNN of $N = 128$ ($\approx$ 17K parameters) , LSTM of $N = 128$ ( $\approx$ 67K parameters), RC uRNN of $N = 512$ ( $\approx$ 7K parameters) , FC uRNN of $N = 128$ ( $\approx$ 33K parameters). The train and test set sizes are chosen to be 100K and 10K respectively. All the models were trained using RMSprop with a learning rate of $1e-3$ and a batch size of 20 or 50 with the best results are being reported here.

The results are presented in figure 2. KRU converges faster than all other baselines even though it has much fewer parameters. This shows the effectiveness of soft unitary constraint which controls the flow of gradients through very long time steps and thus deciding what to forget and remember in an adaptive way. LSTM also converges to the solution and this is achieved through its gating mechanism which controls the flow of the gradients and thus the long term influence. However LSTM has 10 times more parameters than KRU. Both RC uRNN and FC uRNN converges for $T = 100$ but as we can observe, the learning is not stable. The reason for this is that RC uRNN and FC uRNN retains noise since they are strict unitary models. Please note that we do not evaluate RC uRNN for $T = 400$ and $T = 750$ because we found out that the learning is unstable for this model and is often diverging.

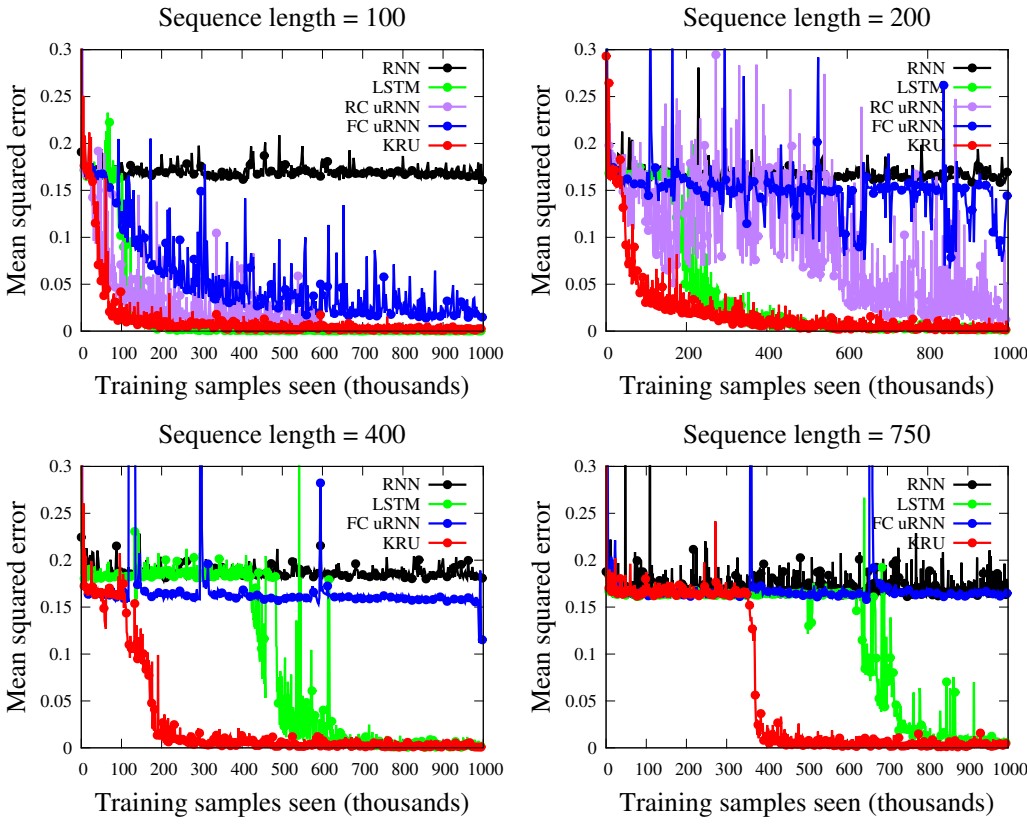

Figure 2: Results on adding problem for $T$=100, $T$=200, $T$=400 and $T$=750. KRU consistently outperforms the baselines on all the settings with fewer parameters.

## 4.3 PIXEL BY PIXEL MNIST

As outlined by Le et al. (2015), we evaluate the Pixel by pixel MNIST task. MNIST digits are shown to the network pixel by pixel and the goal is to predict the class of the digit after seeing all the pixels one by one. We consider two tasks: (1) Pixels are read from left to right from top or bottom and (2) Pixels are randomly permuted before being shown to the network. The sequence length for these tasks is $T = 28 \times 28 = 784$. The size of the MNIST training set is 60K among which we choose 5K as the validation set. The models are trained on the remaining 55K points. The model which gave the best validation accuracy is chosen for test set evaluation. All the models are trained using RMSprop with a learning rate of $1e-3$ and a decay of 0.9.

The results are summarized in figure 3 and table 2. On the unpermuted task LSTM achieve the state of the art performance even though the convergence speed is slow. Recently a low rank plus diagonal gated recurrent unit (LRD GRU) (Barone, 2016) have shown to achieves 94.7 accuracy on permuted MNIST with 41.2K parameters whereas KRU achieves 94.5 with just 12K parameters i.e KRU has 3x parameters less than LRD GRU. Please also note that KRU is a simple model without a gating mechanism. KRU can be straightforwardly plugged into LSTM and GRU to exploit the additional benefits of the gating mechanism which we will show in the next experiments with a KRU-LSTM.

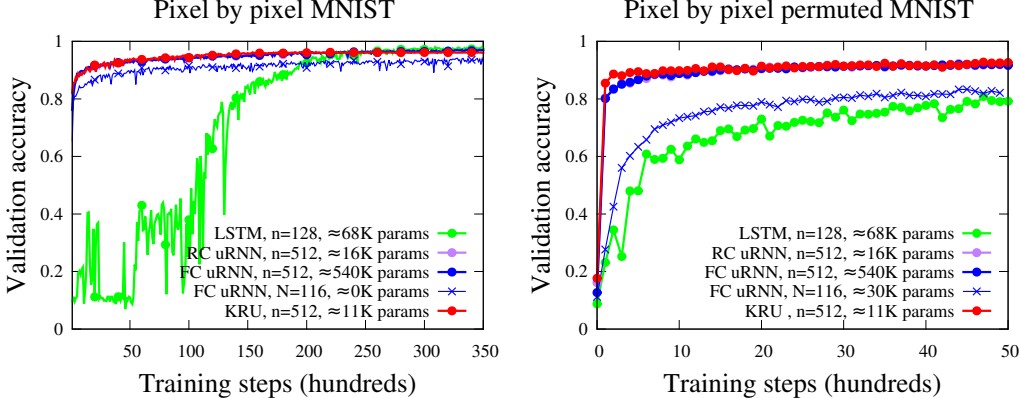

Figure 3: Validation accuracy on pixel by pixel MNIST and permuted MNIST class prediction as the learning progresses.

Table 2: KRU achieves state of the art performance on pixel by pixel permuted MNIST while having up to four orders of magnitude less parameters than other models.

| Model | n | # Parameters | | Unpermuted accuracy | | Permuted accuracy | |
|---|---|---|---|---|---|---|---|
| | | Total | Recurrent | Valid. | Test | Valid. | Test |
| LSTM (Arjovsky et al., 2016) | 128 | ≈68K | ≈65K | 98.1 | **97.8** | 91.7 | 91.3 |
| RC uRNN (Wisdom et al., 2016) | 512 | ≈16K | ≈3.6K | 97.9 | 97.5 | 94.2 | 93.3 |
| FC uRNN (Wisdom et al., 2016) | 512 | ≈540K | ≈524K | 97.5 | 96.9 | 94.7 | 94.1 |
| FC uRNN (Wisdom et al., 2016) | 116 | ≈30K | ≈27K | 92.7 | 92.8 | 92.2 | 92.1 |
| oRNN (Mhammedi et al., 2016) | 256 | ≈11K | ≈8K | 97.0 | 97.2 | - | - |
| EURNN (Jing et al., 2017) | 1024 | ≈13K | ≈4K | - | - | 94.0 | 93.7 |
| KRU | 512 | ≈11K | 72 | 96.6 | 96.4 | 94.7 | **94.5** |

## 4.4 CHARACTER LEVEL LANGUAGE MODELLING ON PENN TREEBANK (PTB)

We now consider character level language modeling on Penn TreeBank data-set (Marcus et al., 1993). Penn TreeBank is composed of 5017K characters in the training set, 393K characters in the validation set and 442K characters in the test set. The size of the vocabulary was limited to 10K most frequently occurring words and the rest of the words are replaced by a special <UNK> character (Mikolov, 2012). The total number of unique characters in the data-set is 50, including the special <UNK> character.

All our models were trained for 50 epochs with a batch size of 50 and using ADAM (Kingma & Ba, 2014). We use a learning rate of $1e-3$ which was found through cross-validation with default beta parameters (Kingma & Ba, 2014). If we do not see an improvement in the validation bits per character (BPC) after each epoch then the learning rate is decreased by 0.30. Back-propagation through time (BPTT) is unrolled for 30 time frames on this task.

We did two sets of experiments to have fair evaluation with the models whose results were available for a particular parameter setting (Mhammedi et al., 2016) and also to see how the performance evolves as the number of parameters are increased. We present our results in table 3. We observe that the strict orthogonal model, oRNN fails to generalize as well as other models even with a high capacity recurrent matrix. KRU and KRU-LSTM performs very close to RNN and LSTM with fewer parameters in the recurrent matrix. Please recall that the computational bottleneck in RNN is the computation of hidden states 2.1 and thus having fewer parameters in the recurrent matrix can significantly reduce the training and inference time.

Recently HyperNetworks (Ha et al., 2016) have shown to achieve the state of the art performance of 1.265 and 1.219 BPC on the PTB test set with 4.91 and 14.41 million parameters respectively. This is respectively 13 and 38 times more parameters than the KRU-LSTM model which achieves 1.47 test BPC. Also Recurrent Highway Networks (RHN) (Zilly et al., 2016) proved to be a promising model for learning very deep recurrent neural networks. Running experiments, and in particular exploring meta-parameters with models of that size, requires unfortunately computational means beyond what was at our disposal for this work. However, there is no reason that the consistent behavior and improvement observed on the other reference baselines would not generalize to that type of large-scale models.

Table 3: Performance in BPC of KRU variants and other models for character level language modeling on Penn TreeBank data-set. KRU has fewer parameters in the recurrent matrix which significantly bring down training and inference time.

| Model | N | # Parameters | | Valid. BPC | Test BPC |
|---|---|---|---|---|---|
| | | Total | Recurrent | | |
| RNN | 300 | $\approx$120K | 90K | 1.65 | 1.60 |
| LSTM | 150 | $\approx$127K | 90K | 1.63 | 1.59 |
| oRNN (Mhammedi et al., 2016) | 512 | $\approx$183K | $\approx$130K | 1.73 | 1.68 |
| KRU | 411 | $\approx$120K | $\approx$38K | 1.65 | 1.60 |
| RNN | 600 | $\approx$420K | 360K | 1.56 | 1.51 |
| LSTM | 300 | $\approx$435K | 360K | **1.50** | **1.45** |
| KRU | 993 | $\approx$418K | $\approx$220K | 1.53 | 1.48 |
| KRU-LSTM | 500 | $\approx$377K | $\approx$250K | 1.53 | 1.47 |

## 4.5 POLYPHONIC MUSIC MODELING

We exactly follow the experimental framework of Chung et al. (2014) for Polyphonic music modeling (Boulanger-Lewandowski et al., 2012) on two datasets: JSB Chorales and Piano-midi. Similar to (Chung et al., 2014) our main objective here is to have a fair evaluation of different recurrent neural networks. We took the baseline RNN and LSTM models of (Chung et al., 2014) whose model sizes were chosen to be small enough to avoid overfitting. We choose the model size of KRU and KRU-LSTM in such way that it has fewer parameters compared to the baselines. As we can in the table 4 both our models (KRU and KRU-LSTM) overfit less and generalizes better. We also present the wall-clock running time of different methods in the figure 4.

Table 4: Average negative log-likelihood of KRU and KRU-LSTM compared to the baseline models.

| Model | n | # Parameters | | JSB Chorales | | Piano-midi | |
|---|---|---|---|---|---|---|---|
| | | Total | Recurrent | Train | Test | Train | Test |
| RNN (Chung et al., 2014) | 100 | $\approx$20K | 10K | 8.82 | 9.10 | 5.64 | 9.03 |
| LSTM (Chung et al., 2014) | 36 | $\approx$20K | $\approx$5.1K | 8.15 | 8.67 | 6.49 | 9.03 |
| KRU | 100 | $\approx$10K | 58 | 7.90 | 8.59 | 7.57 | 8.28 |
| KRU-LSTM | 45 | $\approx$19K | 172 | 7.47 | **8.54** | 7.55 | **8.18** |

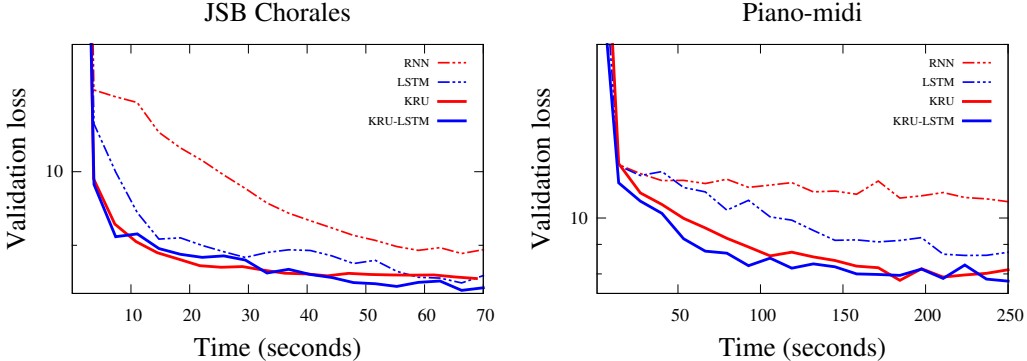

Figure 4: Wall clock training time on JSB Chorales and Piano-midi data-set.

## 4.6 FRAMEWISE PHONEME CLASSIFICATION ON TIMIT

Framewise phoneme classification (Graves & Schmidhuber, 2005) is the problem of classifying the phoneme corresponding to a sound frame. We evaluate the models for this task on the real world TIMIT data-set (Garofolo et al., 1993). TIMIT contains a training set of 3696 utterances among which we use 184 as the validation set. The test set is composed of 1344 utterances. We extract 12 Mel-Frequency Cepstrum Coefficients (MFCC) (Mermelstein, 1976) from 26 filter banks and also the log energy per frame. We also concatenate the first derivative, resulting in a feature descriptor of dimension 26 per frame. The frame size is chosen to be 10ms and the window size is 25ms.

The number of time steps to which back-propagation through time (BPTT) is unrolled corresponds to the length of each sequence. Since each sequence is of different length this implies that for each sample BPTT steps are different. All the models are trained for 20 epochs with a batch size of 1 using ADAM with default beta parameters (Kingma & Ba, 2014). The learning rate was cross-validated for each of the models from $\eta \in \{1e-2, 1e-3, 1e-4\}$ and the best results are reported here. The best learning rate for all the models was found out to be $1e-3$ for all the models. Again if we do not observe a decrease in the validation error after each epoch, we decrease the learning rate by a factor of $\gamma \in \{1e-1, 2e-1, 3e-1\}$ which is again cross-validated. Figure 5 summarizes our results.

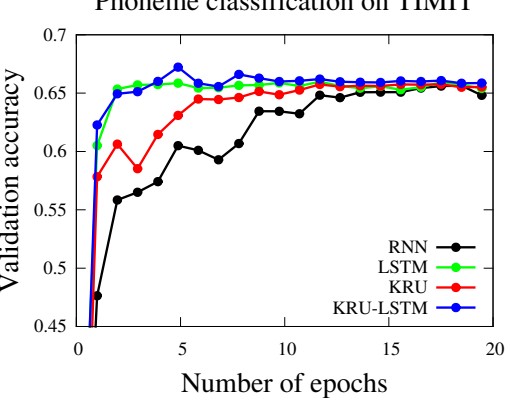

| Model | N | # Parameters Total | # Parameters Recurrent | Valid. accuracy | Test accuracy |
|---|---|---|---|---|---|
| RNN | 600 | ≈406K | 360K | 65.84 | 64.53 |
| LSTM | 300 | ≈406K | 360K | 65.99 | 64.56 |
| KRU | 2048 | ≈195K | 16K | 65.91 | 64.55 |
| KRU-LSTM | 2048 | ≈404K | 66K | **66.54** | **64.81** |

Figure 5: KRU and KRU-LSTM performs better than the baseline models with far less parameters in the recurrent weight matrix on the challenging TIMIT data-set (Garofolo et al., 1993). This significantly bring down the training and inference time of RNNs. Both LSTM and KRU-LSTM converged within 5 epochs whereas RNN and KRU took 20 epochs. A similar result was obtained by (Graves & Schmidhuber, 2005) using RNN and LSTM with 4 times less parameters respectively than our models. However in their work the LSTM took 20 epochs to converge and the RNN took 70 epochs. We have also experimented with the same model size as that of (Graves & Schmidhuber, 2005) and have obtained very similar results as in the table but at the expense of longer training times.

## 4.7 INFLUENCE OF SOFT UNITARY CONSTRAINTS

Here we study the properties of soft unitary constraints on KRU. We use Polyphonic music modeling data-sets (Boulanger-Lewandowski et al., 2012): JSB Chorales and Piano-midi, as well as TIMIT data-set for this set of experiments. We varied the amplitude of soft unitary constraints from $1e - 7$ to $1e - 1$, the higher the amplitude the closer the recurrent matrix will be to the unitary set. All other hyper-parameters, such as the learning rate and the model size are fixed. We present our studies in the figure 6. As we increase the amplitude we can see that the recurrent matrix is getting better conditioned and the spectral norm or the spectral radius is approaching towards 1. As we can see that the validation performance can be improved using this simple soft unitary constraints. For JSB Chorales the best validation performance is achieved at an amplitude of $1e - 2$, whereas for Piano-midi it is at $1e - 1$.

For TIMIT phoneme recognition problem, the best validation error is achieved at $1e - 5$ but as we increase the amplitude further, the performance drops. This might be explained by a vanishing long-term influence that has to be forgotten. Our model achieve this by cross-validating the amplitude of soft unitary constraints. These experiments also reveals the problems of strict unitary models such as RC uRNN (Arjovsky et al., 2016), FC uRNN (Wisdom et al., 2016), oRNN (Mhammedi et al., 2016) and EURNN (Jing et al., 2017) that they suffer from the retention of noise from a vanishing long term influence and thus fails to generalize.

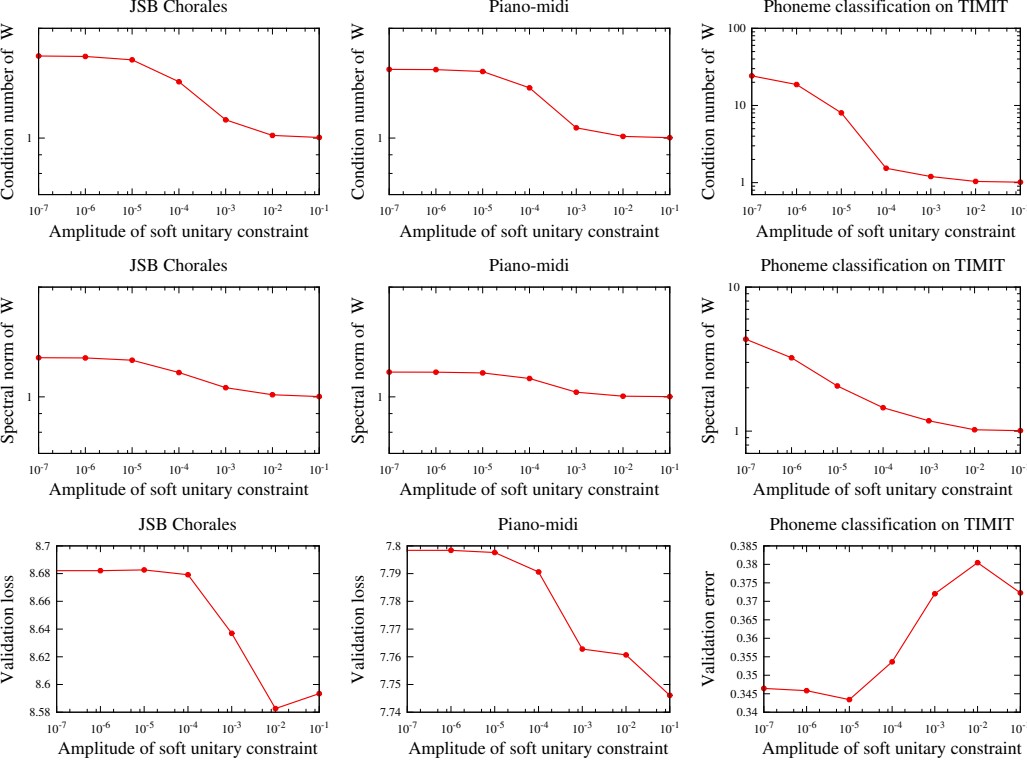

Figure 6: Analysis of soft unitary constraints on three data-sets. First, second and the third column presents JSB Chorales, Piano-midi and TIMIT data-sets respectively.

A popular heuristic strategy to avoid exploding gradients in RNNs and thereby making their training robust and stable is gradient clipping. Most of the state of the art RNN models use gradient clipping for training. Please note that we are not using gradient clipping with KRU. Our soft unitary constraints offer a principled alternative to gradient clipping.

Moreover Hardt et al. (2016) recently showed that gradient descent converges to the global optimizer of linear recurrent neural networks even though the learning problem is non-convex. The necessary condition for the global convergence guarantee requires that the spectral norm of recurrent matrix

is bounded by 1. This seminal theoretical result also inspires to use regularizers which control the spectral norm of the recurrent matrix, such as the soft unitary constraints.

## 5 CONCLUSION

We have presented a new recurrent neural network model based on its core a Kronecker factored recurrent matrix. Our core reason for using a Kronecker factored recurrent matrix stems from it's elegant algebraic and spectral properties. Kronecker matrices are neither low-rank nor block-diagonal but it is multi-scale like the FFT matrix. Kronecker factorization provides a fine control over the model capacity and it's algebraic properties enable us to design fast matrix multiplication algorithms. It's spectral properties allow us to efficiently enforce constraints like positive semi-definitivity, unitarity and stochasticity. As we have shown, we used the spectral properties to efficiently enforce a soft unitary constraint.

Experimental results show that our approach out-perform classical methods which uses $\mathcal{O}(N^2)$ parameters in the recurrent matrix. Maybe as important, these experiments show that both on toy problems (§ 4.1 and 4.2), and on real ones (§ 4.3, 4.4, , and § 4.6), while existing methods require tens of thousands of parameters in the recurrent matrix, competitive or better than state-of-the-art performance can be achieved with far less parameters in the recurrent weight matrix. These surprising results provide a new and counter-intuitive perspective on desirable memory-capable architectures: the state should remain of high dimension to allow the use of high-capacity networks to encode the input into the internal state, and to extract the predicted value, but the recurrent dynamic itself can, and should, be implemented with a low-capacity model.

From a practical standpoint, the core idea in our method is applicable not only to vanilla recurrent neural networks and LSTMS as we showed, but also to a variety of machine learning models such as feed-forward networks (Zhou et al., 2015), random projections and boosting weak learners. Our future work encompasses exploring other machine learning models and on dynamically increasing the capacity of the models on the fly during training to have a perfect balance between computational efficiency and sample complexity.

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

APPENDICES

A  ANALYSIS OF VANISHING AND EXPLODING GRADIENTS IN RNN

Given a sequence of $T$ input vectors: $\mathbf{x}_0, \mathbf{x}_1, \ldots, \mathbf{x}_{T-1}$, let us consider the operation at the hidden layer $t$ of a recurrent neural network:

$$\mathbf{z}_t = \mathbf{W}_t \mathbf{h}_{t-1} + \mathbf{U}_t \mathbf{x}_t + \mathbf{b} \tag{8}$$
$$\mathbf{h}_t = \sigma(\mathbf{z}_t) \tag{9}$$

By the chain rule,

$$\frac{\partial \mathcal{L}}{\partial \mathbf{h}_t} = \frac{\partial \mathcal{L}}{\partial \mathbf{h}_T} \frac{\partial \mathbf{h}_T}{\partial \mathbf{h}_t} \tag{10}$$

$$= \frac{\partial \mathcal{L}}{\partial \mathbf{h}_T} \prod_{k=t}^{T-1} \frac{\partial \mathbf{h}_{k+1}}{\partial \mathbf{h}_k} = \frac{\partial \mathcal{L}}{\partial \mathbf{h}_T} \prod_{k=t}^{T-1} \mathbf{J}_{k+1} \mathbf{W}^T \tag{11}$$

where $\sigma$ is the non-linear activation function and $\mathbf{J}_{k+1} = diag(\sigma^{'}(\mathbf{z}_{k+1}))$ is the Jacobian matrix of the non-linear activation function.

$$\left\| \frac{\partial \mathcal{L}}{\partial \mathbf{h}_t} \right\| = \left\| \frac{\partial \mathcal{L}}{\partial \mathbf{h}_T} \prod_{k=t}^{T-1} \mathbf{J}_{k+1} \mathbf{W}^T \right\| \tag{12}$$

$$\leq \left\| \frac{\partial \mathcal{L}}{\partial \mathbf{h}_T} \right\| \prod_{k=t}^{T-1} \left\| \mathbf{J}_{k+1} \mathbf{W}^T \right\| \tag{13}$$

$$\leq \left\| \frac{\partial \mathcal{L}}{\partial \mathbf{h}_T} \right\| \| \mathbf{W} \|^{T-t} \prod_{k=t}^{T-1} \| \mathbf{J}_{k+1} \| \tag{14}$$

From equation 14 it is clear the norm of the gradient is exponentially dependent upon two factors along the time horizon:

- The norm of the Jacobian matrix of the non-linear activation function $\| \mathbf{J}_{k+1} \|$.
- The norm of the hidden to hidden weight matrix $\| \mathbf{W} \|$.

These two factors are causing the vanishing and exploding gradient problem.

Since the gradient of the standard non-linear activation functions such as tanh and ReLU are bounded between [0, 1], $\| \mathbf{J}_{k+1} \|$ does not contribute to the exploding gradient problem but it can still cause vanishing gradient problem.

B  LONG SHORT-TERM MEMORY (LSTM) (HOCHREITER & SCHMIDHUBER, 1997)

LSTM networks presented an elegant solution to the vanishing and exploding gradients through the introduction of gating mechanism. Apart from the standard hidden state in RNN, LSTM introduced one more state called cell state $c_t$. LSTM has three different gates whose functionality is described as follows:

- Forget gate ($\mathbf{W}_f, \mathbf{U}_f, \mathbf{b}_f$):Decides what information to keep and erase from the previous cell state.
- Input gate ($\mathbf{W}_i, \mathbf{U}_f, \mathbf{b}_i$): Decides what new information should be added to the cell state.
- Output gate ($\mathbf{W}_o, \mathbf{U}_o, \mathbf{b}_o$):Decides which information from the cell state is going to the output.

In addition to the gates, LSTM prepares candidates for the information from the input gate that might get added to the cell state through the action of input gate. Let's denote the parameters describing the function that prepares this candidate information as $\mathbf{W}_c, \mathbf{U}_c, \mathbf{b}_c$.

Given a sequence of $T$ input vectors: $\mathbf{x}_0, \mathbf{x}_1, \ldots, \mathbf{x}_{T-1}$, at a time step $t$ LSTM performs the following:

$$\mathbf{f}_t = \sigma(\mathbf{W}_f \mathbf{h}_{t-1} + \mathbf{U}_f \mathbf{x}_t + \mathbf{b}_f) \tag{15}$$
$$\mathbf{i}_t = \sigma(\mathbf{W}_i \mathbf{h}_{t-1} + \mathbf{U}_i \mathbf{x}_t + \mathbf{b}_i) \tag{16}$$
$$\mathbf{o}_t = \sigma(\mathbf{W}_o \mathbf{h}_{t-1} + \mathbf{U}_o \mathbf{x}_t + \mathbf{b}_o) \tag{17}$$
$$\hat{\mathbf{c}}_t = \tau(\mathbf{W}_c \mathbf{h}_{t-1} + \mathbf{U}_c \mathbf{x}_t + \mathbf{b}_c) \tag{18}$$
$$\mathbf{c}_t = \mathbf{c}_{t-1} \odot \mathbf{f}_t + \hat{\mathbf{c}}_t \odot \mathbf{i}_t \tag{19}$$
$$\mathbf{h}_t = \tau(\mathbf{c}_t) \odot \mathbf{o}_t \tag{20}$$

where $\sigma(.)$ and $\tau(.)$ are the point-wise sigmoid and tanh functions. $\odot$ indicates element-wise multiplication. The first three are gating operations and the 4th one prepares the candidate information. The 5th operation updates the cell-state and finally in the 6th operation the output gate decided what to go into the current hidden state.

## C   UNITARY EVOLUTION RNN (ARJOVSKY ET AL., 2016)

Unitary evolution RNN (uRNN) proposed to solve the vanishing and exploding gradients through a unitary recurrent matrix, which is for the form:

$$\mathbf{W} = \mathbf{D}_3 \mathbf{R}_2 \mathscr{F}^{-1} \mathbf{D}_2 \Pi \mathbf{R}_1 \mathscr{F} \mathbf{D}_1. \tag{21}$$

Where:

- $\mathbf{D}_1, \mathbf{D}_2, \mathbf{D}_3$: Diagonal matrices whose diagonal entries are of the from $\mathbf{D}_{kk} = e^{i\theta_k}$, implies each matrix have $N$ parameters, $(\theta_0, \ldots, \theta_{N-1})$.
- $\mathscr{F}$ and $\mathscr{F}^{-1}$: Fast Fourier operator and inverse fast Fourier operator respectively.
- $\mathbf{R}_1, \mathbf{R}_2$: Householder reflections. $R = \mathbf{I} - 2\frac{\mathbf{v}\mathbf{v}^H}{\|\mathbf{v}\|}$, where $v \in \mathbb{C}^N$.

The total number of parameters for this uRNN operator is $7N$ and the matrix vector can be done $N log(N)$ time. It is parameter efficient and fast but not flexible and suffers from the retention of noise and difficulty in optimization due its unitarity.

## D   FULL CAPACITY UNITARY RNN (WISDOM ET AL., 2016)

Full capacity unitary RNN (FC uRNN) does optimization on the full unitary set instead on a subset like uRNN. That is FC uRNN's recurrent matrix $\mathbf{W} \in U(N)$. There are several challenges in optimization over unitary manifold especially when combined with stochastic gradient method. The primary challenge being the optimization cost is $\mathcal{O}(N^3)$ per step.

## E   ORTHOGONAL RNN (MHAMMEDI ET AL., 2016)

Orthogonal RNN (oRNN) parametrizes the recurrent matrices using Householder reflections.

$$\mathbf{W} = \mathcal{H}_N(\mathbf{v}_N)...\mathcal{H}_{N-K+1}(\mathbf{v}_{N-k+1}). \tag{22}$$

where

$$\mathcal{H}_K(\mathbf{v}_K) = \begin{bmatrix} \mathbf{I}_{N-K} & 0 \\ 0 & \mathbf{I}_K - 2\frac{\mathbf{v}_K \mathbf{v}_K^H}{\|\mathbf{v}_K\|} \end{bmatrix} \tag{23}$$

and

$$\mathcal{H}_1(\mathbf{v}) = \begin{bmatrix} \mathbf{I}_{N-1} & 0 \\ 0 & \mathbf{v} \in \{-1,1\} \end{bmatrix} \tag{24}$$

where $\mathbf{v}_K \in \mathbb{R}^K$. The number of parameters in this parametrization is $\mathcal{O}(NK)$. When $N = K = 1$ and $v = 1$, it spans the rotation subset and when $v = -1$, it spans the full reflection subset.

## F    PROPERTIES OF KRONECKER MATRIX (VAN LOAN, 2000)

Consider a matrix $\mathbf{W} \in \mathbb{C}^{N \times N}$ factorized as a Kronecker product of $F$ matrices $\mathbf{W}_0, \ldots, \mathbf{W}_{F-1}$,

$$\mathbf{W} = \mathbf{W}_0 \otimes \cdots \otimes \mathbf{W}_{F-1} = \otimes_{i=0}^{F-1} \mathbf{W}_i. \tag{25}$$

Where each $\mathbf{W}_i \in \mathbb{C}^{P_i \times Q_i}$ respectively and $\prod_{i=0}^{f-1} P_i = \prod_{i=0}^{F-1} Q_i = N$. $\mathbf{W}_i$'s are called as Kronecker factors.

If the factors $\mathbf{W}_i$'s are $\left\{\begin{array}{l}\text{Nonsingular}\\ \text{Symmetric}\\ \text{Stochatsic}\\ \text{Orthogonal}\\ \text{Unitary}\\ \text{PSD}\\ \text{Toeplitz}\end{array}\right\}$ then $\mathbf{W}$ is $\left\{\begin{array}{l}\text{Nonsingular}\\ \text{Symmetric}\\ \text{Stochatsic}\\ \text{Orthogonal}\\ \text{Unitary}\\ \text{PSD}\\ \text{Block Toeplitz}\end{array}\right\}$

**Theorem 1.** *If $\forall i \in 0, \ldots, F-1$, $\mathbf{W}_i$ is unitary then $\mathbf{W}$ is also unitary.*

*Proof.*

$$\mathbf{W}^H \mathbf{W} = (\mathbf{W}_0 \otimes \cdots \otimes \mathbf{W}_{f-1})^H (\mathbf{W}_0 \otimes \cdots \otimes \mathbf{W}_{f-1}) \tag{26}$$

$$= (\mathbf{W}_0^H \otimes \cdots \otimes \mathbf{W}_{f-1}^H)(\mathbf{W}_0 \otimes \cdots \otimes \mathbf{W}_{f-1}) \tag{27}$$

$$= \mathbf{W}_0^H \mathbf{W}_0 \otimes \cdots \otimes \mathbf{W}_{f-1}^H \mathbf{W}_{f-1} = \mathbf{I}. \tag{28}$$

$\square$

## G    PRODUCT BETWEEN A DENSE MATRIX AND A KRONECKER MATRIX

For simplicity here we use real number notations. Consider a dense matrix $\mathbf{X} \in \mathbb{R}^{M \times K}$ and a Kronecker factored matrix $\mathbf{W} \in \mathbb{R}^{N \times K}$. That is $\mathbf{W} = \otimes_{f=0}^{F-1} \mathbf{W}_f$, where each $\mathbf{W}_f \in \mathbb{R}^{P_f \times Q_f}$ respectively and $\prod_{f=0}^{F-1} P_f = N$ and $\prod_{f=0}^{F-1} Q_f = K$. Let us illustrate the matrix product $\mathbf{X}\mathbf{W}^T$ resulting in a matrix $\mathbf{Y} \in \mathbb{R}^{M \times N}$.

$$\mathbf{Y} = \mathbf{X}\mathbf{W}^T. \tag{29}$$

The computational complexity first expanding the Kronecker factored matrix and then computing the matrix product is $\mathcal{O}(MNK)$. This can be reduced by exploiting the recursive definition of Kronecker matrices. For examples when $N = K$ and $\forall_f \{P_f = Q_f = 2\}$, the matrix product can be computed in $\mathcal{O}(MN \log N)$ time instead of $\mathcal{O}(MN^2)$.

The matrix product in 29 can be recursively defined as

$$\mathbf{Y} = (\ldots (\mathbf{X} \odot \mathbf{W}_0^T) \otimes \cdots \otimes \mathbf{W}_{F-1}^T). \tag{30}$$

Please note that the binary operator $\odot$ is not the standard matrix multiplication operator but instead it denotes a strided matrix multiplication. The stride is computed according to the algebra of Kronecker matrices. Let us define $\mathbf{Y}$ recursively:

$$\mathbf{Y}_0 = \mathbf{X} \odot \mathbf{W}_0 \tag{31}$$

$$\mathbf{Y}_f = \mathbf{Y}_{f-1} \odot \mathbf{W}_f. \tag{32}$$

Combining equation 34 and 32

$$\mathbf{Y} = \mathbf{Y}_{F-1} = (\ldots (\mathbf{X} \odot \mathbf{W}_0^T) \otimes \cdots \otimes \mathbf{W}_{F-1}^T). \tag{33}$$

We use the above notation for $\mathbf{Y}$ in the algorithm. That is the algorithm illustrated here will cache all the intermediate outputs $(\mathbf{Y}_0, \ldots, \mathbf{Y}_{F-1})$ instead of just $\mathbf{Y}_{F-1}$. These intermediate outputs are then

later to compute the gradients during the back-propagation. This cache will save some computation during the back-propagation. If the model is just being used for inference then the algorithm can the organized in such a way that we do not need to cache the intermediate outputs and thus save memory.

Algorithm for computing the product between a dense matrix and a Kronecker factored matrix 34 is given below 1. All the matrices are assumed to be stored in row major order. For simplicity the algorithm is illustrated in a serial fashion. Please note the lines 4 to 15 except lines 9-11 can be trivially parallelized as it writes to independent memory locations. The GPU implementation exploits this fact.

---

**Algorithm 1** Dense matrix product with a Kronecker matrix, $\mathbf{Y} = (\ldots (\mathbf{X}\mathbf{W}_0^T) \otimes \cdots \otimes \mathbf{W}_{F-1}^T)$

---

**Input:** Dense matrix $\mathbf{X} \in \mathbb{R}^{M \times K}$, Kronecker factors $\{\mathbf{W}_0, \ldots, \mathbf{W}_{F-1}\} : \mathbf{W}_f \in \mathbb{R}^{p_f \times q_f}$, Size of
   each Kronecker factors $\{(P_0, Q_0), \ldots, (P_{F-1}, Q_{F-1})\} : \prod_{f=0}^{F-1} P_f = N, \prod_{f=0}^{F-1} Q_f = K$,
**Output:** Output matrix $Y_{F-1} \in \mathbb{R}^{M \times N}$
  1: **for** $f = 0$ to $F - 1$ **do**
  2:   $stride = K/Q_f$
  3:   $index = 0$
  4:   **for** $m = 0$ to $M - 1$ **do**
  5:     $\mathbf{X}_m = \mathbf{X} + m \times K$
  6:     **for** $p = 0$ to $P_f - 1$ **do**
  7:       **for** $s = 0$ to $stride - 1$ **do**
  8:         $\mathbf{Y}_f[index] = 0$
  9:         **for** $q = 0$ to $Q_k - 1$ **do**
 10:           $\mathbf{Y}_f[index] = \mathbf{Y}_f[index] + \mathbf{X}_m[q \times stride + s] \times \mathbf{W}_f[p \times Q_f + q]$
 11:         **end for**
 12:         $index = index + 1$
 13:       **end for**
 14:     **end for**
 15:   **end for**
 16:   $K = stride$
 17:   $M = M \times P_f$
 18:   $X = Y_f$
 19: **end for**

---

## H    GRADIENT COMPUTATION IN A KRONECKER LAYER

Following the notations from the above section G, here we illustrate the algorithm for computing the gradients in a Kronecker layer. To be clear and concrete the Kronecker layer does the following computation in the forward pass 32.

$$\mathbf{Y} = \mathbf{Y}_{F-1} = (\ldots (\mathbf{X} \odot \mathbf{W}_0^T) \otimes \cdots \otimes \mathbf{W}_{F-1}^T). \tag{34}$$

That is, the Kronecker layer is parametrized by a Kronecker factored matrix $\mathbf{W} = \otimes_{f=0}^{F-1} \mathbf{W}_f$ stored as it factors $\{\mathbf{W}_0, \ldots, \mathbf{W}_{F-1}\}$ and it takes an input $\mathbf{X}$ and produces output $\mathbf{Y} = \mathbf{Y}_{F-1}$ using the algorithm 1.

The following algorithm 2 computes the Gradient of the Kronecker factors: $\{\mathbf{gW}_0, \ldots, \mathbf{gW}_{F-1}\}$ and the Jacobian of the input matrix $\mathbf{gX}$ given the Jacobian of the output matrix: $\mathbf{gY} = \mathbf{gY}_{F-1}$.

---

**Algorithm 2** Gradient computation in a Kronecker layer.

---

**Input:** Input matrix $\mathbf{X} \in \mathbb{R}^{M \times K}$, Kronecker factors $\{\mathbf{W}_0, \dots, \mathbf{W}_{F-1}\} : \mathbf{W}_f \in \mathbb{R}^{p_f \times q_f}$, Size of each Kronecker factors $\{(P_0, Q_0), \dots, (P_{F-1}, Q_{F-1})\} : \prod_{f=0}^{F-1} P_f = N, \prod_{f=0}^{F-1} Q_f = K$, All intermediate output matrices from the forward pass: $\{\mathbf{Y}_0, \dots, \mathbf{Y}_{F-1}\}$, Jacobian of output matrix: $\mathbf{gY}_{F-1} \in \mathbb{R}^{M \times N}$
**Output:** Gradient of Kronecker factors: $\{\mathbf{gW}_0, \dots, \mathbf{gW}_{F-1}\}$ and Jacobian of input matrix: $\mathbf{gX} \in \mathbb{R}^{M \times N}$.

1: $T = M \times N$
2: $strideP = 1$
3: $strideQ = 1$
4: $\mathbf{gY} = \mathbf{gY}_{F-1}$
5: **for** $f = F - 1$ to $0$ **do**
6:    $R = strideP \times P_f$
7:    $S = strideQ \times Q_f$
8:    $T = T/P_f$
9:    $\mathbf{Z} = nullptr$
10:    $\mathbf{gZ} = nullptr$
11:    **if** $f == 0$ **then**
12:      $\mathbf{Z} = \mathbf{X}$
13:      $\mathbf{gZ} = \mathbf{gX}$
14:    **else**
15:      $\mathbf{gZ} = \mathbf{Y}_{f-1}$
16:      $\mathbf{Z} = \mathbf{gZ}$
17:    **end if**
18:    $index = 0$
19:    **for** $t = 0$ to $T - 1$ **do**
20:      $\mathbf{Z}_t = \mathbf{Z} + t \times S$
21:      **for** $p = 0$ to $P_f - 1$ **do**
22:        **for** $s = 0$ to $strideQ - 1$ **do**
23:          **for** $q = 0$ to $Q_k - 1$ **do**
24:           $\mathbf{gW}_f[p \times Q_k + 1] = \mathbf{gW}_f[p \times Q_k + 1] + \mathbf{Z}_t[q \times strideQ + s] \times \mathbf{gY}[index]$
25:          **end for**
26:          $index = index + 1$
27:        **end for**
28:      **end for**
29:    **end for**
30:    $index = 0$
31:    **for** $t = 0$ to $T - 1$ **do**
32:      $\mathbf{gY}_t = \mathbf{gY} + t \times R$
33:      **for** $p = 0$ to $P_f - 1$ **do**
34:        **for** $s = 0$ to $strideQ - 1$ **do**
35:          $\mathbf{gZ}[index] = 0$
36:          **for** $q = 0$ to $Q_k - 1$ **do**
37:           $\mathbf{gZ}[index] = \mathbf{gZ}[index] + \mathbf{gY}[q \times strideQ + s] \times \mathbf{W}_f[q \times P_f + q]$
38:          **end for**
39:          $index = index + 1$
40:        **end for**
41:      **end for**
42:    **end for**
43:    $\mathbf{gY} = \mathbf{gZ}$    //We reuse the memory for the intermediate outputs to store the gradients.
44:    $strideQ = S$
45:    $strideP = R \times Q_f/P_f$
46: **end for**

---

