# OpenReview forum: "Kronecker Recurrent Units"
_ICLR.cc/2018/Conference — Invite to Workshop Track_

### Official Review · AnonReviewer2 · 2017-11-22
**Nice idea, more exploration is needed**

**Rating:** 6
**Confidence:** 5

**Review:**

The paper presents a method to parametrize unitary matrices in an RNN as a Kronecker product of smaller matrices. Given N inputs and output, this method allows one to specify a linear transformation with O(log(N)) parameters, and perform a forward and backward pass in O(Nlog(N)) time.
In addition a relaxation is performed allowing each constituent to deviate a bit from unitarity (“soft unitary constraint”).
The paper shows nice results on a number of small tasks.

The idea is original to the best of my knowledge and is presented clearly.
I especially like the idea of “soft unitary constraint” which can be applied very efficiently in this factorized setup. I think this is the main contribution of this work.

However the paper in its current form has a number of problems:

- The authors state that a major shortcoming of previous (efficient) unitary RNN methods is the lack of ability to span the entire space of unitary matrices. This method presents a family that can span the entire space, but the efficient parts of this family (which give the promised speedup) only span a tiny fraction of it, as they require only O(log(N)) params to specify an O(N^2) unitary matrix. Indeed in the experimental section only those members are tested.

- Another claim that is made is that complex numbers are key, and again the argument is the need to span the entire space of unitary matrices, but the same comment still hold - that is not the space this work is really dealing with, and no experimental evidence is provided that using complex numbers was really needed.

- In the experimental section an emphasis is made as to how small the number of recurrent params are, but at the same time the input/output projections are very large, leaving the reader wondering if the workload simply shifted from the RNN to the projections. This needs to be addressed.

- Another aspect of the previous points is that it’s not clear if stacking KRU layers will work well. This is important as stacking LSTMs is a common practice. Efficient KRU span a restricted subspace whose elements might not compose into structures that are expressive enough. One way to overcome this potential problem is to add projection matrices between layers that will do some mixing, but this will blow the number of parameters. This needs to be explored.

- The authors claim that the soft unitary constraint was key for the success of the network, yet no details are provided as to how this constraint was applied, and no analysis was made for its significance.

---

### Official Review · AnonReviewer1 · 2017-11-27
**Interesting idea, but weak experimental setup**

**Rating:** 5
**Confidence:** 4

**Review:**


Summary of the paper
-------------------------------

This paper proposes to factorize the hidden-to-hidden matrix of RNNs into a Kronecker product of small matrices, thus reducing the number of parameters, without reducing the size of the hidden vector. They also propose to use a soft unitary constraint on those small matrices (which is equivalent to a soft unitary constraint on the Kronecker product of those matrices), that is fast to compute. They evaluate their model on 6 small scale RNN experiments.

Clarity, Significance and Correctness
--------------------------------------------------

Clarity: The main idea is clearly motivated and presented, but the experiment section failed to convince me (see details below).

Significance: The idea of using factorization for RNNs is not particularly novel. However, it is really nice to be able to decouple the hidden size and the number of recurrent parameters in a simple way. Also, the combination of Kronecker product and soft unitary constraint is really interesting.

Correctness: There are minor flaws. Some of the baselines seems to perform poorly, and some comparisons with the baselines seems unfair (see the questions below).

Questions
--------------

1. Section 3: You say that you can vary 'pf' and 'qf' to set the trade-off between computational budget and performances. Have you run some experiments where you vary those parameters?
2. Section 4: Are you using the soft unitary constraint in your experiments? Do you have an hyper-parameter that sets the amplitude of the constraint? If yes, what is its value? Are you using it also on the vanilla RNN or the LSTM?
3. Section 4.1: You say that you don't train the recurrent matrix in the KRU version. Do you also not train the recurrent matrix in the other models (RNN, LSTM,...)? If yes, how do you explain the differences? If no, I don't see how those curves compare.
4. Section 4.3: Why does your LSTM in pMNIST performs so poorly? There are way better curves reported in the literature (eg in "Unitary Evolution Recurrent Neural Netwkrs" or "Recurrent Batch Normalization").
5. General: How does your method compares with other factorization approaches, such as in "Factorization Tricks for LSTM Networks"?
6. Section 4: How does the KRU compares to the other parametrizations, in term of wall-clock time?

Remarks
------------

The main claim of the paper is that RNN are over-parametrized and take a long time to train (which I both agree with), but you didn't convinced me that your parametrization solve any of those problems. I would suggest to:
1. Compare more clearly setups where you fix the hidden size.
2. Compare more clearly setups where you fix the number of parameters.
With systematic comparisons like that, it would be easier to understand where the gains in performances are coming from.
3. Add an experiment where you vary 'pf' and 'qf' (and keep the hidden size fixed) to show how the optimization/generalization performances can be tweaked.
4. Add computation time (wall-clock) for all the experiments, to see how it compares in practice (this could definitively weight in your favor, since you seems to have a nice CUDA implementation).
5. Present results on larger-scale applications (Text8, Teaching Machines to Read and Comprehend, 3 layers LSTM speech recognition setup on TIMIT, DRAW, Machine Translation, ...), especially because your method is really easy to plug in any existing code available online.

Typos / Form
------------------

1. sct 1, par 3: "using Householder reflection vectors, it allows a fine-grained" -> "using Householder reflection vectors, which allows a fine-grained"
2. sct 1, par 3: "This work called as Efficient" -> "This work, called Efficient"
5. sct 1, par 5: "At the heart of KRU is the use of Kronecker" -> "At the heart of KRU, we use Kronecker"
6. sct 1, par 5: "Thanks to the properties of Kronecker matrices" -> "Thanks to the properties of the Kronecker product"
7. sct 1, par 5: "vanilla real space RNN" -> "vanilla RNN"
8. sct 2, par 1: "Consider a standard recurrent" -> "Consider a standard vanilla recurrent"
9. sct 2, par 1: "step t RNN" -> "step t, a vanilla RNN"
11. sct 2.1, par 1: "U and V, this is efficient using modern BLAS" -> "U and V, which can be efficiently computed using modern BLAS"
12. sct 2.3, par 2: "matrices have a determinant of 1 or −1, i.e., the set of all rotations and reflections respectively" -> "matrices, i.e., the set of all rotations and reflections, have a determinant of 1 or −1."
13. sct 3, par 1: "are called as Kronecker" -> "are called Kronecker"
14. sct 3, par 3: "used it's spectral" -> "used their spectral"
15. sct 3, par 3: "Kronecker matrices" -> "Kronecker products"
18. sct 4.4, par 3: "parameters are increased" -> "parameters increases"
19. sct 5: There is some more typos in the conclusion ("it's" -> "its")
20. Some plots are hard to read / interpret, mostly because of the round "ticks" you use on the curves. I suggest you remove them everywhere. Also, in the adding problem, it would be cleaner if you down-sampled a bit the curves (as they are super noisy). In pixel by pixel MNIST, some of the legends might have some typos (FC uRNN), and you should use "N" instead of "n" to be consistent with the notation of the paper.
21. Appendix A to E are not necessary, since they are from the literature.
22. sct 3.1, par 2: "is approximately unitary." -> "is approximately unitary (cf Appendix F)."
23. sct 4, par 1: "and backward operations." -> "and backward operations (cf Appendix G and H)."

Pros
------

1. Nice Idea that allows to decouple the hidden size with the number of hidden-to-hidden parameters.
2. Cheap soft unitary constraint
3. Efficient CUDA implementation (not experimentally verified)

Cons
-------

1. Some experimental setups are unfair, and some other could be clearer
2. Only small scale experiments (although this factorization has huge potential on larger scale experiments)
3. No wall-clock time that show the speed of the proposed parametrization.

---

### Official Review · AnonReviewer3 · 2017-11-27
**A nice idea, well presented**

**Rating:** 7
**Confidence:** 3

**Review:**

Typical recurrent neural networks suffer from over-paramterization. Additionally, standard RNNs (non-gated versions) have an ill-conditioned recurrent weight matrix, leading to vanishing/exploding gradients during training. This paper suggests to factorize the recurrent weight matrix as a Kronecker product of matrices. Additionally, in order to avoid vanishing/exploding gradients in standard RNNs, a soft unitary constraint is used. The regularizer is specifically nice in this setting, as it suffices to have the Kronecker factors be unitary. In the empirical section, several RNNs are trained using this approach, using only ~ 100 recurrent parameters, and still achieve comparable results to state-of-the-art approaches. The paper argues that the recurrent state should be high-dimensional (in order to be able to encode the input and extract predictive information) but the recurrent dynamic should be realized by a low-capacity model.

Quality: The paper is well written.

Clarity: Main ideas are clearly presented.

Originality/Significance: Kronecker factorization was introduced for Convolutional networks (citation is in the paper). Soft unitary constraints also have been introduced in earlier work (citations are also in the paper). Nevertheless, showing that these two ideas work also for RNNs in combination (and seeing, e.g. the nice relationship between Kronecker factors and unitary) is a relevant contribution. Additionally, this approach allows a significant reduction of training time it seems.

---

### Decision · Program_Chairs · 2018-01-29
**ICLR 2018 Conference Acceptance Decision**

**Decision:**

Invite to Workshop Track

**Comment:**

I tend to agree with the most positive reviewer who characterizes the work with the following statements:

"Kronecker factorization was introduced for Convolutional networks (citation is in the paper). Soft unitary constraints also have been introduced in earlier work (citations are also in the paper). Nevertheless, showing that these two ideas work also for RNNs in combination (and seeing, e.g. the nice relationship between Kronecker factors and unitary) is a relevant contribution."

The most negative reviewer feels that the experimental work could have evaluated the different components explored here more clearly. For this reason the AC recommends an invitation to the workshop track.